# A Study of Ralex Membrane Morphology by SEM

**DOI:** 10.3390/membranes9120169

**Published:** 2019-12-06

**Authors:** Elmara M. Akberova, Vera I. Vasil’eva, Victor I. Zabolotsky, Lubos Novak

**Affiliations:** 1Department of Analytical Chemistry, Voronezh State University, Universitetskaya pl.1, 394018 Voronezh, Russia; 2Department of Physical Chemistry, Kuban State University, ul. Stavropolskaya 149, 350040 Krasnodar, Russia; 3MEGA a.s., Pod Vinicí 87, 471 27 Stráž podRalskem, Czech Republic

**Keywords:** ralex ion-exchange membrane, microscopic analysis, structural inhomogeneity

## Abstract

A comparative analysis of the effect of the manufacturing technology of heterogeneousion-exchange membranes Ralex CM Pes manufactured by MEGA a.s. (Czech Republic) on the structural properties of their surface and cross section by SEM was carried out. The CM Pes membrane is a composite of a sulfonated ion-exchanger with inert binder of polyethylene and reinforcing polyester fiber. In the manufacture of membranes Ralex the influence of two factors was investigated. First, the time of ion-exchange grain millingvaried at a constant resin/polyethylene ratio. Second, the ratio of the cation-exchanger and the inert binder of polyethylene varied. It has been found that the membrane surface becomes more electrically homogeneous with the growth of the ion-exchanger loading and a decrease in its particle size. With an increase in the milling time of resin grainsfrom 5 to 80 min a more than 1.5-fold decrease in their radius and in the distance between them was revealed.Besides, there is a 1.5-fold decrease in the fraction, as well as in the size of pores and structure defects. The fraction of the ion-exchange phase on the membrane surface decreases by 7%. With an increase in the resin loading from 45 to 70 wt %, the growth of the fraction of conducting regions on the surface is almost twofold, while their sizes remain practically unchanged. More significant changes in the surface structure of the studied membranes are established in comparison with the cross section. An increase in the resin content in the membranes from 45 to 70 wt % corresponds to a 43% increment of its fraction on the cross-section.The increase in the ion-exchanger content of Ralex membranes is accompanied by the growth of the fraction of macropores and structure defects on the membrane surface by 70% and a twofold decrease in the distance between conducting zones.

## 1. Introduction

In the electrochemical behavior of heterogeneous ion-exchange membranes, the electrical inhomogeneity of their surface, consisting ofthe presence of the regions with high (ion-exchanger phase) and low (polyethylene phase) conductivity, plays an important role. In [1,2,3,4,5,6,7,8,9,10], the possibility of mass transfer intensification in an electromembrane system by improving the morphology of the surface of ion-exchange membranes was demonstrated. The use of membranes with optimized surface morphology in the process of electrodialysis for desalting and deionization of natural waters and technological solutions creates the prerequisites for a significant increase in the efficiency of these processes in the limiting and overlimiting current modes.

Objective information about membranestructure is provided by visualization methods: high and low vacuum scanning electron microscopy [1,3,11,12,13,14,15,16,17,18], scanning force microscopy [11,15,19] and optical microscopy [20,21]. SEM with energy dispersive X-ray analysis is a very useful tool for qualitative and semiquantitative analysis of the elemental composition of the surface or cross-section of membranes of various types under different operating conditions [14,20,22,23,24,25].

Ion-exchange membranes are operational only in the swollen state. However, the traditionally used methods of atomic force microscopy and a high-vacuum version of scanning electron microscopy make it possible to examine samples only in a dry state. The application of optical microscopy to the study of swollen membranes is complicated by a close light conductivity of ion-exchange regions and polyethylene, which negatively affects the accuracy of determining the size of conducting andinert regions [26]. In [3,27], a method was proposed and tested for determining the fraction of ion-conducting and inert phases on the surface of swollen heterogeneous membranes based on the SEM results obtained by scanning dry samples. The expression for recalculating the fraction of the conducting surface of the dry membrane with respect to the swollen sample was obtained under the assumption that the increase in the linear dimensions of the membranes in the process of swelling occurs mainly due to an increase in the size of the ion exchange material. A low-vacuum SEM allows visualizing of the surface and cross-section structure of ion-exchange membranes of various types in the swollen state [9,16,28,29].

The experimental information about the morphology of the surface of ion-exchange membranes, the dimensions of conducting and inert regions is necessary for understanding the mechanisms of intensification of electromembrane processes, including a theoretical description of the transfer of electrolyte ions through ion-exchange membranes with a heterogeneous surface [3,30]. The idea of surface modifying of homogeneous and heterogeneous ion-exchange membranes has been widely developed in order to enhance the mass transfer in overlimiting modes due to the intensification of heteroelectroconvective instability [31,32]. Another way to obtain membranes with a dominant electroconvective ion transfer mechanism is to modify the surface of famous commercial heterogeneous membranes by changing their manufacturing technology. For example, by varying the resin particle size or changing the resin/inert binder ratio, a compromise between electrochemical and mechanical properties can be achieved to obtain a better heterogeneous membrane [33]. The aim of the work was an experimental estimation by SEM of the effect of the technology of manufacturing heterogeneoussulfonated membranes Ralex on the structural properties of their surface and cross-section.

## 2. Experimental

### 2.1. Membranes

Experimental samples of heterogeneous ion-exchange membranes Ralex CM Pes (“MEGA” a.s., Straz pod Ralskem, Czech Republic) of various manufacturing technologies were selected for the study.The membranes of the first series were made from themilled resin (58 wt %) of varying sizes and polyethylene (42 wt %), polyester fiber was used as the reinforcing material [34]. The sizes of the cation-exchanger were varied by using milling time from 5 to 80 min. The experimental samples of the second series of Ralex CM Pes membranes differ in the resin/inert binder ratio. In the manufacture of membranes of this batch, the resin loading varied from 45 to 70 wt %.

Prior to experiments, the membrane samples were treated with NaCl solutions with different concentrations [10].

### 2.2. Membrane Characterization

The total static exchange capacity of the membranes (*Q*) was determined by acid-base titration.Thewatercontent (*W*) represented the ratio of the mass of water in the swollen sample to its mass. The thickness of the samples (*b*) was measured with a micrometer, and the density of ion-exchange membranes (*d*) by a pycnometric method [35].

### 2.3. Scanning Electron Microscopy

The study of the morphology of the membrane surface was performed by scanning electron microscopy (SEM) using a microscope JSM-6380 LV (JEOL, Tokyo, Japan). In the chamber with the sample under study, the pressure was adjustable, which allowed investigating membranes in the low-vacuum mode in the working (swollen) state [16,28]. Electron-microscopic photography of the samples was carried out in the low-vacuum mode using backscattered electrons at an accelerating voltage of 20 kV. Preliminary preparation of membrane samples was not made since the main advantage of low-vacuum devices is the neutralization of the surface charge by positively charged gas atoms, which made the coating with a conducting layer unnecessary [36]. The main factor affecting the number of backscattered electrons is the elemental composition of the detecting region [37]. Therefore, the light areas in the image correspond to the regions of the analyzed surface containing atoms with higher atomic number due to a more intense reflection of electrons. Thus, the areas of light gray color correspond to the sections of the ion-exchanger, whose ionic groups contain sulfur, oxygen and sodium atoms. The dark gray color corresponds to polyethylene, which contains carbon atoms. The dark border between them indicates the presence of a pore or a defect in the structure.

The information about the chemical composition of the membrane surface was obtained by mapping the elemental composition using an energy dispersive spectrometer. The color coding for X-ray spectrum imaging allowed combining data on several elements in one image.In the obtained images (Figure 1), areas painted in red color indicate the presence of sulfur. They correspond to regions of the ion-exchanger containing fixed $SO_{3}^{2-}$-groups. Inert regions of polyethylene, which mainly contain carbon atoms, corresponding to a blue color.

### 2.4. Software

Quantitative assessment of structural characteristics on the membrane surface and cross-section was carried out using the authors’ software package. The software package is an application developed in the Delphi programming language [38]. The application provides automation of the analysis of the surface morphology and cross-section of ion-exchange membranes by means of digital processing of SEM images. The software package provides a wide range of possibilities for digital image processing: (a) various types of filters (median and averaging) are proposed for noise reduction; (b) brightness jumps are highlighted using the Sobel, Roberts, and Laplacian gradients; (c) image binarization is performed using a global or local threshold. To assess the relative fraction of microphases on the membrane surface, the method of growing regions was used. Besides, in the software package, an additional module for constructing a histogram of the distribution of conducting surface sections along effective radii is included.

For the determination of the proportion and radiusof the microphasessimultaneously, taking into account the “size effect” of the inhomogeneous surface of heterogeneous ion-exchange membranes, the value for relative magnification was chosen. At magnifications of 100–500, both the fraction of the ion-exchanger and the fraction of pores on the membrane surface remain almost unchanged (Figure 2). At magnification 1200 these characteristics increase by about two times. Therefore, to calculate the proportion of microphases, it is preferable to use magnifications from 100 to 500. When determining the radii of microphases, on the contrary, the values are constant with magnification of more than 500. This is due to the fact that, at small magnifications, small ion-exchangers and pores are not visualized. Therefore, to calculate the radii of microphases, it is preferable to use SEM images with magnification of 500–1200. Taking into account the “size effect” to determine the proportion and size of microphases simultaneously the recommended magnification is 500.

The fraction of ion-exchanger was determined by the formula *S* = (*ΣS_i_/S*) · 100%, where *ΣS_i_* is the total area of ion-exchange regions, *S* is the area of the scanned region. To calculate the radius of the ion-exchange region *R* the program simulated circular sections which are as were equivalent to the real areas of the ion-exchangers of an arbitrary shape (Figure 3). The weighted average radius of microphases was calculated according to [16].

The fraction *P* and the weighted average pore radius $\bar{r}$ on the membrane surface were calculated in the same way. Under the concept of a “pore” the area between the ion-exchange particle and polyethylene was assumed. As pores, were also taken into account the cracks on polyethylenedue to the fact that they can be through.Membrane structure is porous, and thus the ion transport can proceed either by direct contact of ion-exchanger particles or by short channels filled up with the electrolyte [13]. Using a magnification of 500 at microscopic imaging allows us to determine only the macroporous structure of the membranes.

The processing of the measurement results was carried out by the methods of mathematical statistics [39]. When determining thesurface and cross-section characteristics ofmembranes using the SEM method, various parts of each membrane sample were scanned 8–10 times.

## 3. Results and Discussion

### 3.1. Surface Morphology of Heterogeneous Membranes with Different Resin Particle Sizes

The surface structure of the investigated heterogeneous ion-exchange membranes Ralex is inhomogeneous (Figure 4). Light areas of ion-exchanger particles and dark areas of inert polyethylene are visualized. At the same time, a significant part of the membrane surface is shielded with polyethylene, and the conducting phase, in which ionic groups are localized, occupies a smaller fraction. It can be seen that the conducting regions on the surface of the heterogeneous CM Pes membranes have different shapes and they are arrangedrandomly.

It has been established that with the growth of the time ofresin milling, the fraction of ion-exchangers on the surface of the swollen membranes decreases by 7% (Figure 5a). Earlier [40,41] it was found that the resin fractionon the surface of commercially manufactured “MEGA” a.s. Ralex membranes in the swollen state is 25–30%.

The analysis of the ion-exchangerradius distribution is shown in Figure 6. The more milling time of resin particles, the greater total amount of them on the surface. However, the maximum on the distribution curve for Ralex CM Pes membrane samples remains constant and is in the region of 1–2 μm. According to the estimates made in [40], the characteristic radius of the conducting regions on the surface of commercially manufactured by MEGA a.s. swollen samples of CM Pes cation-exchange membranes range from 0.5 to 7 μm. Similar data were obtained byhigh vacuum scanning electron microscopyfor driedsamples at magnification of 200 in [42]: SEM picture analysis of the membrane sample Ralex CM has shown that the size of conductive polymer granules varies in range from units to tens of microns.

An increase in the milling time of the ion-exchanger corresponds to a decrease in the weighted average radius $\bar{R}$ of resin particles on the surface of swollen membranes by 20%. For the membranes with ion-exchanger after 80 min of milling $\bar{R}=1.52\pm 0.03$ µm (Figure 5b). It is well-known [40] that the value of the weighted average radius for the commercially manufactured Ralex cation-exchange membrane is $\bar{R}=1.88$ µm.

With an increase in the milling time, both the effective radius of the resin particles and the distance between them decrease (Figure 5b). Linear correlation $\bar{l}=9.608\bar{R}-9.318$ (correlation coefficient *r* = 0.864) between the ion-exchanger particle size $\bar{R}$ and the distance between them $\bar{l}$ was established.The decrease in the size of ion-conducting sections on the surface of the membranes Ralex is accompanied by a 1.5-fold decrease in the fraction of macropores and structure defects. With an increase in the milling time of ion-exchange particles, a significant increase in the fraction of macropores with a radius of less than 0.5 μm was found (Figure 6b). At the same time, the weighted average radii of macropores are 1.55 ± 0.03 and 1.27 ± 0.04 μm for membranes with ion-exchanger after 5 and 80 min of milling, respectively (Figure 5b).

In Figure 7 the effect of the time of resin particlemilling on the values of relative changes of physicochemical parameters is shown. An analysis of the characteristics indicates that with an increase in the milling time from 5 to 80 min, the ion-exchange capacity per gramof dry membrane decreases by 7%, the water content increases by 9%. For the studied membranes, the difference inthe thickness of the swollen samples is 23%. The decrease in the ion-exchange capacity with the growth ofmilling time is due to the stronger manifestation of the effect of capsulation [43,44] of smaller ion-exchange particles with polyethylene during manufacturing. Confirmation is a decrease in the resin fraction ofthe membrane surface with an increase in milling time(Figure 5a).

An increase in the time of resin milling also causes a decrease in the fraction of macropores filled with an electrically neutral solution on the membrane surface (Figure 4,5). Apparently, this is accompanied by the growth of the fraction of mesopores and nanopores, which determine the main transport characteristics of membranes, in particular, water content [45]. For example, it is known that the swelling of sulfonated ion-exchange membranes is accompanied by an increase in the content of pores with a radius of 5 to 100 nm [46]. In this case, the content of macropores with a radius of more than 100 nm remains almost unchanged.

### 3.2. Surface Morphology of Heterogeneous Membranes with Different Resin/Inert Binder Ratios

Micrographs of the surfaceand cross-section of membranes with different resin/binder ratios are shown in Figure 8a–f. As can be seen in figures, individual particles are tightly surrounded by the polyethylene.

Analysis of SEM images of membranes with different ratios of resin and inert components indicates more significant changes in the surface structure compared to the bulk.An increase in the resin loading in the membranes from 45 to 70 wt % corresponds to an 80% increase in its fraction on the surface (Figure 9a).At the cross-section of membranes, the increase in the fraction of ion-exchange regions is 43%. The growth of the fraction of the ion-exchange component on the membrane surface is accompanied by an increase in surface porosity by 67% (Figure 9b).

The resin particle size varies from 0.8 to 38 μm (Figure 10), and the diameters of pores and structure defects are 0.8–18 μm. Analysis of the histograms of the ion-exchanger radius distribution (Figure 10) revealed that smaller particles predominate on the surface.The weighted average radius of ion-exchangers and macropores on the membrane surface and cross-section with an increase in the resin content remains almost constant (Figure 9).

With an increase in the fraction of the conducting phase, the weighted average length of the non-conducting regions between the resin particles decreases almost twofold and is 4.9 and 2.5 μm for the samples with the maximum (70 wt %) and minimum (45 wt %) content of the ion-exchanger, respectively.

It was found that with an increase in the ion-exchange resin content from 45 to 70 wt %, the total exchange capacity of Ralex CM membranes changed 1.6 times, and its value in terms of a gram of dry membrane with a maximum ion-exchanger content was 4.23 ± 0.09 (Table 1).

## 4. Conclusions

The SEM method revealed significant differences in the surface and bulkcharacteristics of Ralex heterogeneoussulfonated ion-exchange membranes when their production technology changes. It has been established that the surface of membrane samples becomes electrically more homogeneous both with an increase in the resin content and with an increase in the milling time of the grains (with a decrease in the particle size).

The increase in the milling time of the resin particles from 5 to 80 min in the production of membranes leads to a decrease in radii of ion-exchange regions and distances between them. Thegreater thetime of the ion-exchangegrainmilling, the smaller thefraction and size of macropores and structure defects. At the same time, the fraction of the conducting ion-exchange phase on the surfacedecreases by 7%.

With an increase in the resin loading from 45 to 70 wt %, the growth in the fraction of ion-exchange regions on the membrane surface from 21 to 38% was established. More significant changes in the surface structure of the studied membranes are established in comparison with the cross-section. An increase in the resin content in the membranes from 45 to 70 wt % corresponds to a 43% increase in its fraction on the cross-section.At the same time, the weighted average radius of resin particles and macropores remains constant. A twofold decrement in the size of inert regions of polyethylene between ion-exchangers was revealed.The growth in the loading of ion-exchange resin in heterogeneous Ralex membranes is accompanied by a significant increase in macroporosity and a fraction of structure defects on the surface.

## Figures and Tables

**Figure 1 membranes-09-00169-f001:**
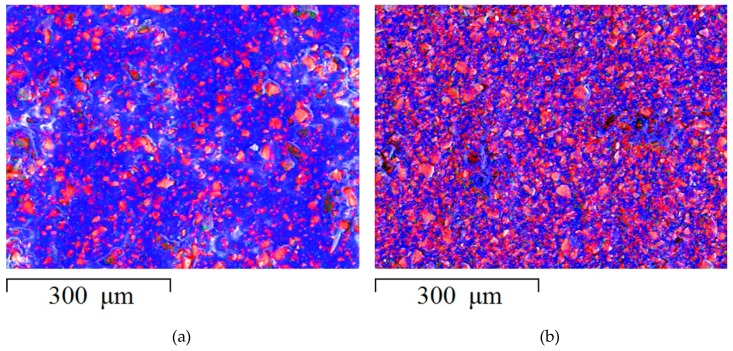
Composite X-ray maps of the S and C elements distribution over the surface of Ralex membranes with resin content (**a**) 45 and (**b**) 70 wt % at ×200 magnification. Element S corresponds to the red color. Element C corresponds to the blue color.

**Figure 2 membranes-09-00169-f002:**
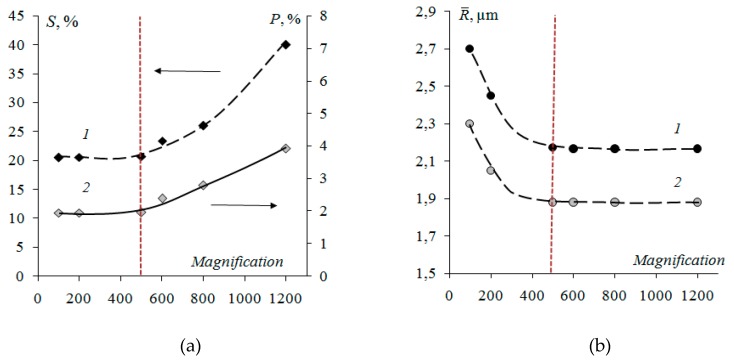
Fraction (**a**) and weighted average radius (**b**) of ion-exchange particles (1), macropores and structure defects (2) on the surfaceof Ralex CM Pes membrane with resin loading 45 wt %.

**Figure 3 membranes-09-00169-f003:**
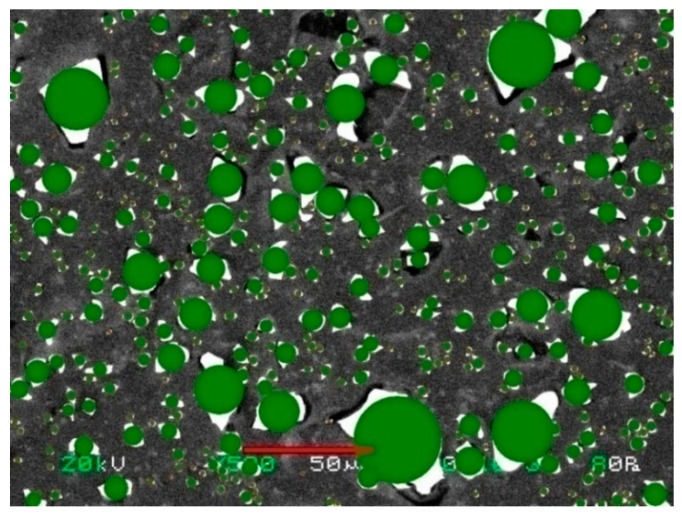
**Figure 3**. Scanning electron microscope (SEM) image of the Ralex membrane surface with software-simulated ion-exchange sections of the circular shape.

**Figure 4 membranes-09-00169-f004:**
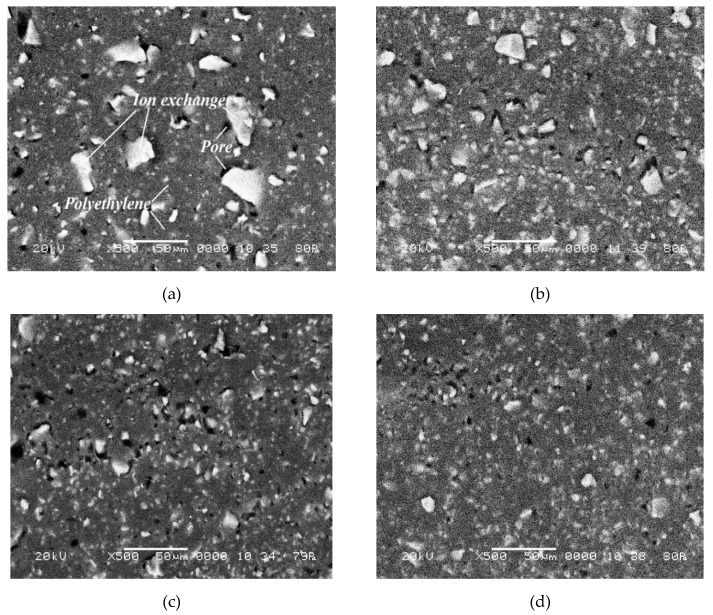
SEM images of the surface of swollen CM Pes membrane samples at a magnification of 500. The time of resin particle milling: 5 (**a**), 40 (**b**), 60 (**c**) and 80 (**d**) min.

**Figure 5 membranes-09-00169-f005:**
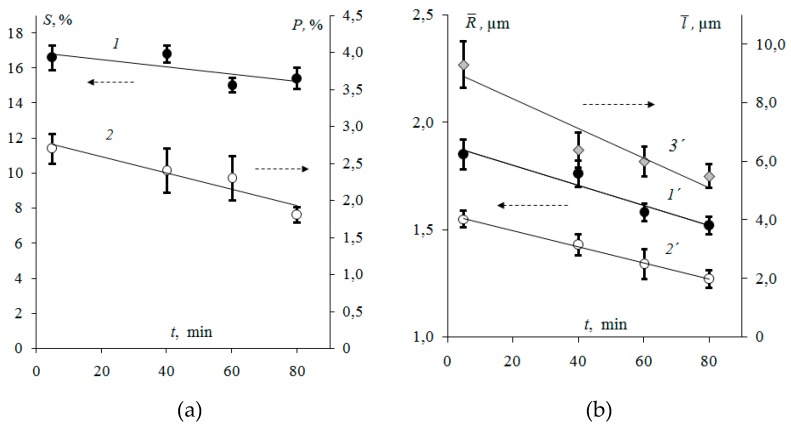
Fraction (**a**) andweighted average radius (**b**) of ion-exchange particles (1,1’), macropores and structure defects (2,2´) and the effective distance between ion-exchangers (3´)on the surface of heterogeneous membranes Ralex with different time of resin particlemilling.

**Figure 6 membranes-09-00169-f006:**
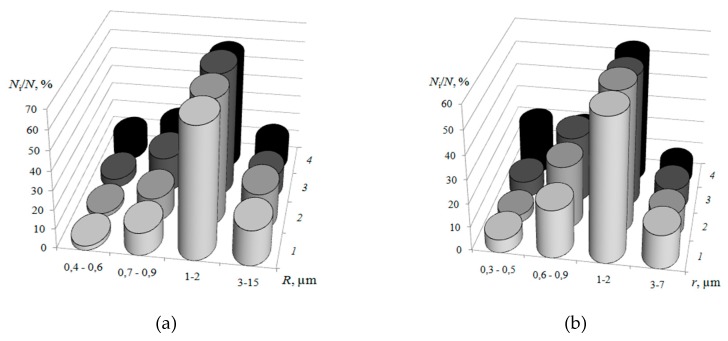
Ion-exchange region (**a**) and macropore (**b**)radius distribution on the surface of swollen samples of CM Pes heterogeneous membranes with ion-exchanger milling time of 5 (1), 40 (2), 60 (3) and 80 (4) min.

**Figure 7 membranes-09-00169-f007:**
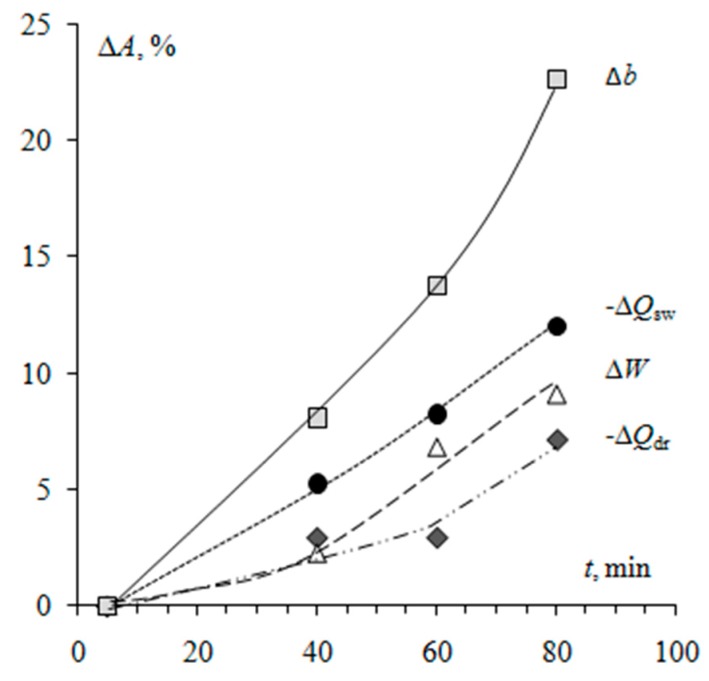
Relative changes in physicochemical characteristics of Ralex membranes with different resin milling time. Relative changes Δ*A*% in physicochemical parametersАwere calculated according to the equation: Δ*A*% = 100(*А_t_*-*А*_5min_)/*А*_5min_, where *А*_5min_—value of a physicochemical characteristic of the membrane with ion-exchanger after milling during 5 min. *Q*_sw_—total exchange capacity per gram of the swollen membrane, mmol/g; *Q*_dr_—total exchange capacity per gram of the dry membrane, mmol/g; *W*—water content, $g_{H_{2}O}/g_{\mathrm{swoll}.\mathrm{membr}}$; *b*—swollen membrane thickness,μm.

**Figure 8 membranes-09-00169-f008:**
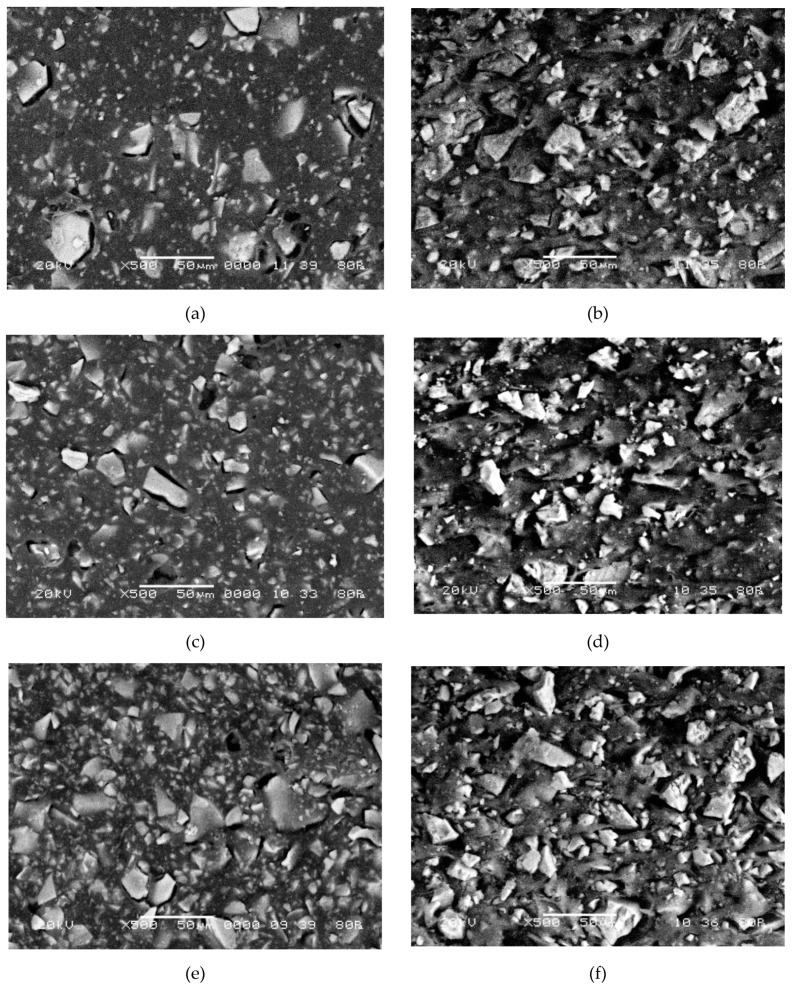
SEM images of the surface (**a**,**c**,**e**) and cross-section (**b**,**d,f**) of swollen CM Pesmembrane at magnification of 500. Ion-exchanger content: 45 (**a**,**b**), 55 (**c**,**d**), and 70 (**e**,**f**) wt %.

**Figure 9 membranes-09-00169-f009:**
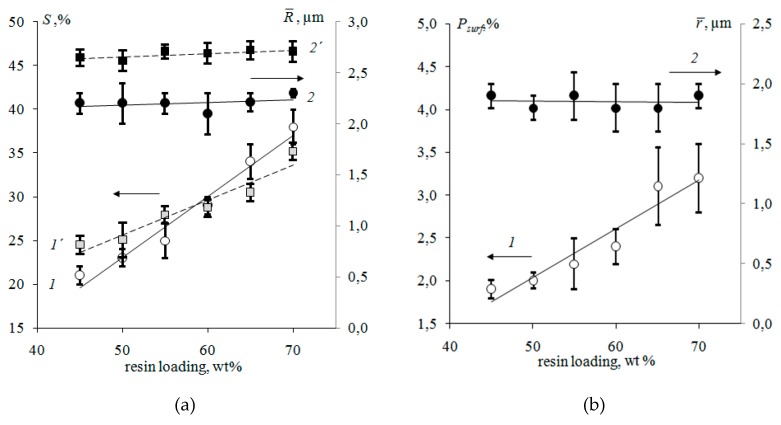
Fraction (1,1´) and weighted average radius (2,2´) of ion-exchange particles (**a**), macropores and structure defects (**b**) on the surface (1,2) and cross-section (1´,2´) of Ralex CM Pes heterogeneous membranes with various resin loading.

**Figure 10 membranes-09-00169-f010:**
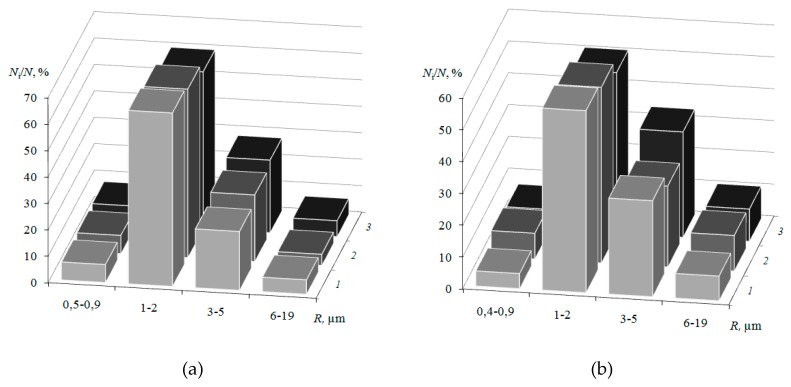
Ion-exchanger radius distribution on the surface (**a**) and cross-section (**b**) of the swollen membranes CM Pes with resin content: 45 (1), 55 (2) and 70 (3) wt %.

**Table 1 membranes-09-00169-t001:** The physicochemical characteristics of Ralex membranes with different resin content.

Resin Content, wt %	*Q*_sw_, mmol/g	*Q*_dr_, mmol/g	$W,\;g_{H_{2}O}{/g}_{\mathrm{swoll}.\mathrm{membr}}$	*b*, μm
45	1.88 ± 0.07	2.66 ± 0.09	0.29	540 ± 20
50	1.90 ± 0.02	2.83 ± 0.02	0.33	550 ± 10
55	1.93 ± 0.05	3.00 ± 0.07	0.36	585 ± 5
60	2.07 ± 0.04	3.38 ± 0.07	0.39	610 ± 5
65	2.16 ± 0.06	3.76 ± 0.09	0.42	655 ± 5
70	2.34 ± 0.05	4.23 ± 0.09	0.45	715 ± 15

The revealed changes in the structure of the surface and cross-section of ion-exchange membranes are responsible for the increase in water content and membrane thickness by 55 and 32%, respectively.

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
