# Peer review of "A Study of Ralex Membrane Morphology by SEM"

_membranes, 2019, doi:10.3390/membranes9120169_

Round 1

Reviewer 1 Report

Review of the manuscript “A study of Ralex membrane morphology by SEM”.

Authors: E. M. Akberova, V. I. Vasil’eva, V. I. Zabolotsky, L. Novak

The separation efficiency of the electro-membrane systems depends on the heterogeneity of the membrane surface. The presence of local electric heterogeneity allows to decrease of the electro-kinetic instability threshold to low value of electric potential drop and to decrease the ionic resistivity. In the case of heterogeneous membrane composed of a mixture of high conductive ion exchanger grains in a low conductive polymeric matrix, the authors have studied the effect of the size distribution and the resin particle loading on the surface heterogeneity. The investigation has been carried out by the analysis of images obtained from low vacuum scanning electron microscopy.

Overall, the authors present original results. However, I feel that the data interpretation must be strengthen and the description of the experimental methodology may be more detailed.

I/ Main remarks:

1/As concerns the milling process: the authors should describe the experimental method. The variation of the ionic exchanger size distribution with the milling time should be displayed. How the grain shape depends on the milling time. Does the properties of the ion exchanger such as the IEC, the ionic conductance is modified by the milling? How this distribution can be compared to the distribution at the membrane surface?

2/ The size and the spatial distribution of the grains at the surface are described. Does the transversal repartition is influenced by the milling process and the resin loading. The term “loading” must be clearly defined (percentage in mass or volume, swollen or dry) because it is surprising that in figure 7 the surface percentage S_surf is twice lower than the resin loading. Why this difference?

3/ Figure 4: the authors show the grain and the macropore size distribution for two milling times. The distribution at t=40 and t=60 min should be displayed. Even if the histogram are interesting, It seems that the grain size and the macropore one are composed of four classes: [0,0.6], [0.7,0.9], [1,2],[2,15] µm for the grains and [0,0.5], [0.6,.9], [1,2],[3,7] µm for the pores. Owing to the high variation of the grain and pore shape, I think that the use of these classes is accurate enough. I am not convinced on the relevance of the maximum at 0.5-0.6 µm in figure 4b. I also think that it should be clearer to use the fraction instead of the number of grains and pores. Different weight can be used: the number, the size, the surface, the specific size (1/r) … . A discussion about these distributions should be added because we do not know to which parameter the electro-kinetic instability is sensitive (and how) and the ionic conductivity depends on the specific surface.

4/ The authors observe a decrease of the effective distance with the milling time. It would be interested to also analyse this distance in function of the resin size. The same analysis can be carried out between two grain classes. Is the spatial distribution isotropic for all the grain sizes? These questions also concern the loading investigation.

5/ The macropores include the hollows against the grains and the cracks (structure defects). What do the authors mean by cracks. These pores have a longitudinal length and their shape is very far away from a disk. So, the cracks should be separated from the hollows in the analysis of the distributions.

6/ Figure 5 and 7: I assume that in figure 7 (and in section 3.2) the grains are not milled and that in figure 5 (and in section 3.1) the loading is 58% if I have understood the paragraph in section 2.1 Figure 5 shows that, at t=0 min, the grain radius and the macropore radius have a value of 1.89 µm and 1.56 µm respectively. Why these values are smaller than the values in figure 7? What is the variation of the surface fraction the grains and the macropores with the time milling?

7/ Figure 8: the distributions can also be divided into four classes: [0,0.9], [1,2] [3,4], [4,10] µm.

II/ Other remarks:

80: does the word “content” mean loading? 108: what is the meaning of S and C. I think that C must be replaced by P. 125: the term “optimum” must be discussed (definition and criterion) 139: the radius r must be replaced by R 152-154: Do the authors used several membranes for each milling time? 162: the term “chaotic” must be clarified.

L.170: The sentence can be modified by : “The analysis of the radius distribution ….”

197: the sentence must be clarified: the macropores gather pores and structure defects. The macropore number is compared to what?

Reviewer 2 Report

In this work authors reported A study of relax membrane morphology by SEM. The results are interesting and sound. It would be suitable to publish in Membranes journal after considering following points.

There are some typo errors in overall manuscript. Authors need to correct them. The English language of the manuscript need to be polished. Does the incorporation of resin particles influence the ion exchange capacity of the particles? Does por size in the membrane influence the ion exchange capacity?

Round 2

Reviewer 1 Report

The manuscript is accepted for publication